# Evaluation of the Interaction between Gum Arabic Addition and Stocking Density on Growth Performance, Carcass Characteristics, and General Health Parameters of Broiler Chickens

**DOI:** 10.3390/ani13193024

**Published:** 2023-09-26

**Authors:** Hani H. Al-Baadani, Rashed A. Alhotan, Mahmoud M. Azzam

**Affiliations:** Department of Animal Production, College of Food and Agriculture Science, King Saud University, P.O. Box 2460, Riyadh 11451, Saudi Arabia; mazzam@ksu.edu.sa

**Keywords:** broiler, performance, blood, carcass, morphology, gum arabic, prebiotic

## Abstract

**Simple Summary:**

Stocking density in broilers (intensive production) is a method to increase meat production and profitability. However, the use of gum arabic as a prebiotic (soluble fiber) is a possible strategy to maintain performance and gut health and prevent physiological stress in this intensive production. Therefore, the aim of this study was to investigate the interaction between the addition of gum arabic and various stocking densities on performance, intestinal morphology, carcass characteristics, lymphoid organs, and selected blood indices of broiler chickens. In this study, gum arabic as a prebiotic was found to improve growth performance, production efficiency, and intestinal morphology, while high stocking density had negative effects on broilers. Further studies are needed to determine the mechanism under various conditions.

**Abstract:**

The present study aims to investigate the interaction between the addition of gum arabic as a prebiotic and various stocking densities on performance indicators, intestinal morphology, carcass characteristics, lymphoid organs, and selected blood indices of broiler chickens. A total of 816 1-day-old male broilers (Ross 308) were used and randomly divided into six blocks as replicates with eight treatments per block (forty-eight floor pens) based on 4 × 2 factorial arrangements with four dietary treatments containing 0.00% (CONT), 0.12% gum arabic (T1), 0.25% gum arabic (T2), and 0.10% commercial prebiotic (T3) and two stocking densities (normal = 28 kg/m^2^; high = 50 kg/m^2^). All performance indicators were evaluated during the feeding phases. Blood biochemical indicators were analyzed at 36 days of age. At 37 days of age, carcass characteristics, lymphoid organs, and intestinal morphology were measured. On days 1–36, growth performance indicators were negatively affected at high stocking density, but all growth performance indicators except feed intake improved in chickens receiving T1–T3 compared to CONT (*p* < 0.05). The relative weight of total small intestine and weight-to-length ratio showed a significant interaction between treatments and stocking density (*p* < 0.05). A high stocking density decreased pre-slaughter weight, carcass weight, and dressing yield, while legs and thymus increased (*p* < 0.05). None of the interactions or treatments affected carcass characteristics or lymphoid organs (*p* > 0.05). Indicators of blood biochemistry were not affected by treatments, stocking density, or their interaction (*p* > 0.05), except for uric acid, creatinine, and aspartate aminotransferase, which were higher at a high stocking density (*p* < 0.05). In conclusion, gum arabic as a prebiotic improved growth performance, production efficiency, and intestinal morphology in broilers. In contrast, high stocking density negatively affected performance, production efficiency, some blood indices, carcass weight, dressing yield, and intestinal morphology. Further research is needed to determine the mechanism.

## 1. Introduction

The modern poultry meat industry relies on intensive production in confined housing systems with large numbers of birds. Such intensive production maximizes meat production while minimizing costs. However, birds raised in this manner may become stressed and susceptible to disease [1,2]. Therefore, many previous studies continuously try to achieve the highest possible stocking density of broilers in the smallest possible floor space (more kg live weight per 1 m^2^) to reduce production costs and increase profitability while maintaining broiler chickens’ performance [3,4,5]. Vargas-Rodrıguez et al. [6] found that stocking density (10 or 16 chickens/m^2^) within the normal range had no effect on broiler performance or health. However, exceeding the stocking density of 28 to 40 kg/m^2^ at the end of the growing season (marketing age) could affect broiler health and performance [7]. High stocking density negatively affects the balance of gut microbiota of broilers and causes dysbiosis [8]. In addition, the morphology of the small intestine of broilers was affected by high stocking density, which reduced their ability to absorb nutrients [9]. Gum arabic is one of the most important natural prebiotics because it mainly contains soluble fiber such as rhamnose, arabinose, and galactose, which have a beneficial effect on the host by stimulating the activity of beneficial bacteria through their ability to ferment in the cecum of birds [10,11]. In addition, the use of soluble fiber, including gum arabic and prebiotics, is one of the possible approaches to maintaining performance and gut health and preventing physiological stress in broiler chickens [12,13,14]. Previous studies have shown that use of gum arabic can effectively improve growth performance, small intestine development, nutrient absorption, and some blood biochemical indicators in broiler chickens [15,16,17,18,19]. Another study found that chickens fed various levels of gum arabic had no effect on carcass characteristics [20].

The hypothesis of the present study is that greater crowding (high stocking density) means a less hygienic environment, and this may be a stress factor for broiler chickens. The additional administration of gum arabic as prebiotic may be beneficial in such conditions to help broilers cope with stress. Therefore, the aim of this study was to investigate the effect of gum arabic on growth performance, small intestine morphology, carcass characteristics, lymphoid organs, and selected blood biochemical indicators of broiler chickens raised in floor pens under various stocking density conditions.

## 2. Materials and Methods

### 2.1. Ethics Approval

The procedures and samples used in the present study were approved by the Ethics Committee of the Deanery of Scientific Research, King Saud University, Kingdom of Saudi Arabia (internal reference number: KSU-SE-20-39).

### 2.2. Feed Additives Used

Gum arabic (*Acacia senegal*) was obtained from an export company in Khartoum State, Sudan. The gum arabic in powder form was chemically analyzed according to the method described by Hani et al. [14]. The prebiotic (*Saccharomyces cerevisiae*) was purchased commercially with a description of the chemical analysis and then used according to the manufacturer’s recommendation. The chemical composition of gum arabic and prebiotic is shown in Table 1.

### 2.3. Study Design, Birds, and Housing

A total of 816 one-day-old male broilers (Ross 308) were housed in floor pens (0.90 m × 0.90 m) based on market weight at 36 days of age in a completely randomized block design with 4 × 2 factorial arrangements with four dietary treatments and two different stocking densities. All chicks were weighed and randomly assigned to six blocks as replicates with eight treatments per block (forty-eight pens). The dietary treatments were as follows: Control (CONT) broilers received a basal diet without additives, while T1 and T2 received a basal diet with 0.12 and 0.25% gum arabic, respectively; T3 received a basal diet with 0.10% inactivated stabilized *Saccharomyces cerevisiae* as a commercial prebiotic (Milan, Italy). Within each dietary treatment, two different stocking densities were performed as follows: a normal stocking density of 28 kg/m^2^ (12 chickens/pen) according to Ross broiler management guidelines in houses with controlled environment and a high stocking density of 50 kg/m^2^ (22 chickens/pen).

The basal diet was formulated on form mash in two feeding stages, starter (1–14 days) and grower (15–36 days), to meet the nutrient requirements of Ross 308 (Aviagen, 2019, New York, USA) as indicated in Table 2. Room temperature (°C) and relative humidity (HR%) were set at 35 °C and 22% HR when the chicks arrived (first day) and then decreased by 2 °C every three days until they were constant at nearly 22 °C and 53% HR after 25 days. The lighting program was offered for 24 h (30–40 lux) until 7 days of age, then 23 h of light (minimum 20 lux) and 1 h of darkness from 8 to 36 days. Feed and water were available to all birds throughout the duration of the study. In addition, there were no differences between stocking density factors in terms of space for feeder and drinker [1]. All chicks were vaccinated against Newcastle disease at the hatchery according to the manufacturer’s instructions (Fort Dodge Animal Health-USA, Overland Park, KS, USA).

### 2.4. General Performance Evaluation 

All birds were weighed on days 1, 14, and 36 during the study period to determine their average body weight. Weight gain and feed intake were calculated for the starter (1–14 days), grower (15–36 days), and the entire stages (1–36 days) according to Diler et al. [21]. Feed conversion ratio was determined based on weight gain and feed intake during the feeding stage [22]. Production efficiency index as a parameter of economic, productive, and welfare status of broiler chickens was evaluated during the feeding stage [23]. The cumulative mortality rate for each dietary treatment was expressed as a percentage during the feeding stages [24].

### 2.5. Small Intestine Morphology

After slaughter (37 days of age), 12 chickens from each treatment were randomly selected to remove the small intestine. The small intestine and its fragments (duodenum, jejunum, and ileum) were measured [25]. The weight and length of duodenum, jejunum, and ileum were expressed as a percentage of the total small intestine. In addition, the weight of the total small intestine (SI) was calculated as a percentage of the live weight. The weight (W) and length (L) of the small intestine were used to calculate the weight-to-length ratio (W/L).

### 2.6. Carcass Characteristics

At 37 days of age, as appropriate to local commercial conditions, 12 chickens from each treatment (2 chickens from each experimental unit (pen) within the dietary treatment) were randomly selected and slaughtered via decapitation after being deprived of feed for 10 h to ensure that the digestive tract was empty. Then, pre-slaughter weight (PSW) was determined using an electronic scale (Adventurer OHAUS, AR3130, Pine Brook, NJ, USA). After slaughter, the carcasses (CW) were weighed. 

The breast (pectoralis major and minor) and legs (thigh and drumstick) were weighed separately and expressed as a percentage of live weight [25]. The percentage of dressing yield (DY) was calculated according to Thema et al. [26]. All lymphoid organs (thymus, bursa, and spleen) were removed and weighed separately to calculate the percentage of live weight [27].

### 2.7. Serum Biochemical Indices

Blood samples (3 mL) were collected at the end of the grower stage (35 days) from 2 chickens per pen (12 chickens per treatment, density) via the wing vein in tubes without heparin. All samples were immediately separated into serum by centrifugation (Thermo Fisher Scientific, Labofuge 200, Dreieich, Germany) at 2500× *g* for 20 min and then stored at −20 °C until analysis of blood biochemical indicators. Blood biochemical indicators include total protein (TP), albumin (ALB), glucose (GLU), triglycerides (TG), cholesterol (CHO), uric acid (UA), creatinine (CREA), alanine aminotransferase (ALT), and aspartate aminotransferase (AST), which were analyzed with colorimetric kits (Randox Laboratories Limited, London, UK) according to the manufacturer’s instructions using an automated spectrophotometric analyzer (Chem Well, Awareness Technology, Palm City, FL, USA).

### 2.8. Statistical Analysis

The pens were used as the experimental unit for all parameters studied, based on a completely randomized block design. Data were statistically analyzed based on a 2 × 4 factorial analysis of variance (two different SD × four TRT) using the general linear models of the Statistical Analysis System software [28]. The statistical model used was as follows: Observed (Y*_ijb_*) = general mean (μ) + stocking density (SD*_i_*; *i* = low, high) + dietary treatments (TRT*_j_*; *j* = CONT, T1, T2, T3) + SD × TRT*_ij_* + block (B*_b_*) + the random error (e*_ijb_*). A normality test (Shapiro–Wilk test) and homogeneity of variances (Levene’s test) were performed before the statistical analysis of the data. Tukey’s test (*p* < 0.05) was used to detect significant differences between means. On the other hand, all relative data were transformed (data/100) to check for significant differences at *p* < 0.05. Finally, all values were expressed as mean ± standard error of the means (SEM) for each parameter.

## 3. Results

### 3.1. Growth Performance Parameters

The effects of dietary treatments and stocking density on growth parameters of broiler chickens are shown in Table 3. The results showed that live weight and weight gain at 14 days and at 1–14 days, respectively, had a significant interaction effect between treatments and stocking density (*p* = 0.014 and *p* = 0.014; respectively). In chickens fed 0.12% gum arabic (T1), body weight and weight gain increased at a normal stocking density and decreased at a high stocking density compared with CONT, T2, and T3. Live weight (36 days) and weight gain (15–36 days and 1–36 days) were lower in chickens at high stocking density compared with normal stocking density (*p* < 0.001). In addition, chickens fed diet supplements (T1–T3) had higher live weight and weight gain than chickens fed CONT (*p* < 0.001), while these values were not affected by the interaction between treatments and stocking density (*p* > 0.05).

For the starter stage (1–14 d), there was a significant interaction effect between stocking density and treatment on feed intake (*p* = 0.008). Feed intake was higher in chickens receiving T1 at normal stocking density, while it decreased at high stocking density compared with the other treatments. On days 15–36 d and 1–36 d, feed intake was decreased at high stocking density compared with normal stocking density (*p* < 0.001). In contrast, feed intake was not affected by the interaction between treatments and stocking density or by the response during this stage (*p* > 0.05).

The effects of dietary treatments with stocking density on feed efficiency and production of broiler chickens are shown in Table 4. The results of the current study showed that a high stocking density improved the feed conversion ratio and production efficiency index on days 1–14, while there was a negative effect on days 15–36 and 1–36 compared to normal stocking density (*p* < 0.001). In addition, chickens fed dietary supplements (T1–T3) increased the production efficiency index and improved the feed conversion ratio on days 15–36 and 1–36 (*p* < 0.001), whereas it had no effect on days 1–14 (*p* > 0.05), compared with CONT. The interaction between treatments and stocking density had no effect on the feed conversion ratio and production efficiency index (*p* > 0.05).

The mortality rate was the same for all dietary treatments and the control group at a high stocking density (one chicken per treatment) during the grower stage (15–36 days old), while there was no mortality in chickens raised at a low stocking density, so no data were published in this study. 

### 3.2. Small Intestine Morphology

The effects of dietary treatments with stocking density on small intestine morphology of broiler chickens are shown in Table 5. The results show that the relative weight and length of small intestinal fragments (duodenum, jejunum, and ileum) were not affected by the treatments, stocking density, or their interaction (*p* > 0.05). The SI ratio and W:L showed a significant interaction effect between treatments and stocking density (*p* = 0.025 and *p* = 0.007; respectively). Chickens fed the diet treatments (T1–T3) had higher W:L and SI under a normal stocking density compared with CONT and a high stocking density (*p* < 0.05), but it had no effect compared with T3.

### 3.3. Carcass Characteristics and Lymphoid Organs

The effects of dietary treatments with stocking density on carcass characteristics and lymphoid organs of broiler chickens are shown in Table 6. The current results show that a high stocking density resulted in a decrease in PSW, CW, DY, and back (*p* < 0.001), while the relative weight of legs and thymus increased compared with normal stocking density (*p* = 0.023 and *p* = 0.001; respectively). The relative weights of breast, bursa, and spleen were not affected by stocking density (*p* > 0.05). The interaction effect or dietary treatments did not affect all parameters of carcass characteristics and lymphoid organs (*p* > 0.05).

### 3.4. Blood Biochemical Indices

The effects of dietary treatments with stocking density on blood biochemistry indicators of broiler chickens are shown in Table 7. The results show that the blood biochemistry indicators including the concentrations of TP, GLU, TG, CHO, UA, CREA, ALT, and AST were not affected by the treatments and the interaction between treatments and stocking density (*p* > 0.05). Albumen concentration (ALB) showed a significant interaction effect between treatments and stocking density (*p* = 0.001). A high stocking density resulted in higher UA, CREA, and AST concentrations compared with a normal stocking density (*p* = 0.003, *p* = 0.008, and *p* = 0.001; respectively). Other blood biochemical indicators (TP, ALB, GLU, TG, CHO, and ALT) had no effect on the stocking density (*p* > 0.05).

## 4. Discussion

Greater crowding (high stocking density) means a less hygienic environment, and this can be a stressor on broiler performance. The additional administration of gum arabic as a prebiotic could be beneficial in such conditions to help broilers cope with stress. Accordingly, this study was conducted to investigate this objective by using two different stocking densities with the addition of gum arabic or prebiotics to broiler diets and measuring their effects on growth performance, intestine morphology, carcass yield, lymphoid organs, and selected blood indices. The current results showed that weight gain and production efficiency increased and feed conversion improved, but there was no difference in feed intake when chickens were raised at high stocking density in the starter stage (1–14 days of age) compared to normal stocking density. This could indicate that high stocking density during this period has a positive effect on chick performance by maintaining thermoregulation, which is incomplete at 1–14 days of age, which could also be due to their smaller size and space requirements. On the other hand, our results showed that body weight, weight gain, feed conversion ratio, and production efficiency were negatively affected when chickens were raised at a high stocking density at 15–36 and 1–36 days of age. These results are consistent with those of Heidari et al. [29] and Miao et al. [30], who confirmed that a high stocking density (18 and 20 chickens/m^2^) at 35 days of age resulted in a decrease in weight gain and feed consumption compared to chickens with normal stocking density. Ghanima et al. [2] reported that a high stocking density has a negative impact on performance indicators of broiler chickens. A high stocking density deteriorates litter and air quality by increasing litter moisture and the volatilization of ammonia, which negatively affects growth performance during the grower stage due to the increasing size of chickens [31]. In addition, the disruption of gut microbial ecology that supports digestion, absorption, and production of beneficial molecules may contribute to poor growth performance of broiler chickens raised at 22–42 and 1–42 days of age at high stocking density [32]. Therefore, high stocking density is one of the stressors that negatively affect growth performance such as body weight, weight gain, and feed intake [9]. Production efficiency and feed conversion ratio are used as indicators of the economic status of broiler production [33]. Thus, a higher production efficiency index and lower feed conversion indicate better performance and feed efficiency when chickens are kept at normal stocking density (28 kg/m^2^).

The current results show that gum arabic and prebiotics (T1–T3) improved growth performance parameters by increasing live weight, weight gain, and production efficiency, while feed intake did not differ during grower and overall stage (15–36 and 1–36 days of age). These results are consistent with those of Khan et al. [34,35], who found that the administration of gum arabic as a prebiotic improved overall performance indicator in broilers. A prebiotic-enriched diet improved weight gain, feed intake, and carcass weight in broilers [36]. Zhen et al. [37] reported that a *Saccharomyces cerevisiae*-derived prebiotic increased the body weight gain and feed intake and improved the feed conversion ratio in broilers. The improvement in the feed conversion ratio via dietary treatments could be attributed to the increase in body weight gain by improving the activity of gut microbiota and feed digestibility. Gum arabic has the ability to ferment through the activity of commensal bacteria in the gut, which could reflect its positive effect on the performance indicators of broilers [38]. In contrast, Tabidi and Ekram [39] reported that gum arabic had no effect on weight gain, feed intake, and feed conversion in broilers. Houshmand et al. [40] showed that the use of mannooligosaccharide derived from the cell wall of *Saccharomyces cerevisiae* as a feed additive had no effect on growth performance parameters of broilers. Morphological measurements of the intestine are used as indicators to evaluate the function of the small intestine and its absorption capacity [41]. Chickens fed T1 to T3 had a higher relative W:L ratio and higher SI compared to CONT at a normal stocking density. However, gum arabic as a natural prebiotic may improve nutrient absorption in broilers [16,42]. This suggests that dietary supplementation improves intestinal morphology, which is reflected in performance parameters and carcass yield by increasing the surface area for nutrient absorption in broilers. The relative weight of SI and the W:L ratio decreased when chickens were kept at a high stocking density compared to a normal stocking density. This suggests that the decrease in the relative weight of the W:L ratio and the total weight of SI leads to a decrease in nutrient uptake ability and thus negatively affects growth performance at a high stocking density during the grower stage [9,43]. 

Selected biochemical indicators in blood are among the tests performed to evaluate the body metabolism of broiler chickens [44]. Therefore, selected blood biochemical indicators were determined to investigate the effects of feed additives and various stocking densities on the health and nutritional status of broiler chickens. Our results showed that the concentrations of TP, GLU, TG, CHO, and ALT were not affected by the treatments, stocking density or their interaction. Singh et al. [45] reported that the biochemical indicators in the blood of broiler chickens were not affected by high stocking density. The gum arabic-fed groups (T1 and T2) had lower ALB concentrations, while the addition of a prebiotic (T3) at a normal stocking density gave a higher concentration of ALB than that in CONT and other treatments at a high stocking density. Normally, a high albumin concentration is one of the physiological stress indicators in broiler chickens [46]. Therefore, the ALB concentration was probably increased in T1 and T2 at a high stocking density as a homeostatic response to stress. A high stocking density resulted in higher UA, CREA, and AST concentrations compared to normal stocking density. These results suggest that some changes in blood biochemical indicators (UA, CREA, and AST) may be due to broiler stocking density. These results are in agreement with those of Ghanima et al. [2] and Nasr et al. [47], who reported that a high stocking density (40 kg/m^2^) at 35 days of age resulted in a decrease in the concentrations of UA, CREA, and AST compared to chickens with a normal stocking density. The high AST activity in broiler chickens kept at a high stocking density could be due to increased competition for feed and water, which increases muscle injury, which is the reason for the higher AST activity in blood [48]. The addition of gum arabic had no effect on the ALT and AST activity of rabbits [19].

Dressing yield and carcass components are important criteria for evaluating the carcass traits and slaughter value of broiler chickens [49]. The results of the current study showed that carcass traits and lymphoid organs were not affected by treatments or by the interaction between treatments and stocking density. In contrast to our results, chickens fed prebiotics had a higher carcass and breast meat yield [50]. Pre-slaughter weight (PSW), CW, and DY decreased, while relative leg and thymus weight increased when chickens were kept at a high stocking density compared to a normal stocking density. Similarly, the results of Cengiz et al. [33] showed that a high stocking density decreased the slaughter weight and dressing yield of broiler chickens. In contrast, several studies showed that stocking density had no effect on carcass characteristics, DY, and lymphoid organs [27,51,52]. Other relative weights of lymphoid organs (bursa and spleen) were not affected by treatments, stocking density, or their interaction. These results do not agree with those of Sato et al. [53], who found that the relative weights of lymphoid organs (bursa and spleen) were higher in broiler chickens fed gum arabic. Dietary supplementation with prebiotics (β-glucan and mannooligosaccharide) resulted in higher relative weights of lymphoid organs [54]. In contrast, Houshmand et al. [40] found that mannooligosaccharide as a prebiotic supplement at rate of 0.10% had no effect on the relative weight of lymphoid organs. 

## 5. Conclusions

In conclusion, the present results provide useful evidence that the use of gum arabic as a prebiotic improves growth performance, production efficiency, and small intestinal morphology without negatively affecting blood biochemical indicators, carcass characteristics, or lymphoid organs to the same extent as the prebiotic (*Saccharomyces cerevisiae*), based on the results of the entire experimental period (1–36 d) in broiler chickens. At the same time, they do not provide a solution for chickens kept at a high stocking density. Higher stocking density (50 kg/m^2^) of broilers negatively affected growth performance, production efficiency, some blood indices (UA, CREA and AST), carcass weight, dressing yield, and intestinal morphology. Further studies are needed to determine the potential mechanism of this supplementation by testing for stress indicators and the intestinal ecosystem. 

## Figures and Tables

**Table 1 animals-13-03024-t001:** Chemical composition of gum arabic and commercial prebiotic on dry matter basis.

Item	Gum Arabic ^1^	Commercial Prebiotic ^2^
Crude Analysis		
Dry matter	90.68	92.00
Crude protein	2.30	28.90
Lipid	0.10	2.57
Ash	5.13	7.8
Mannanoligosacharide	-	16.00
β-glucan	-	18.00
Rhamnose	8.40	-
Arabinose	26.00	-
Galactose	40.18	-
Glucuronic acid	18.23	-
Amino acid Analysis		
Threonine	0.14	1.06
Glutamic acid	0.17	1.11
Valine	0.13	1.21
Isoleucine	0.03	1.10
Leucine	0.17	1.66
Tyrosine	0.06	0.88
Phenylalanine	0.12	0.93
Histidine	0.15	0.64
Lysine	0.06	1.39
Arginine	0.03	1.02
Cysteine	0.10	0.39
Mineral Analysis		
Calcium	1.10	0.23
Phosphorus	0.60	3.6
Sodium	0.02	0.50
Magnesium	0.46	0.90
Zinc	0.0002	1.5
Iron	0.75	0.71

^1^ The chemical composition analysis of gum arabic was performed according to Al-Baadani et al. [14]. ^2^ The chemical composition analysis of the commercial prebiotic (*Saccharomyces cerevisiae*) was obtained by the manufacturer of the product.

**Table 2 animals-13-03024-t002:** Basal diet components and nutrients content on a dry matter basis.

Basal Diet Components, %	Feeding Stages	
	Starter (1–14 Days)	Grower (15–36 Days)
Corn	53.84	55.50
Soybean meal (48% CP)	38.24	35.90
Soy oil	3.43	4.65
Monocalcium phosphate	1.59	1.41
Limestone	1.54	1.39
DL-Methionine	0.37	0.31
L-Lysine HCL	0.25	0.15
L-Threonine	0.16	0.10
Common salt	0.38	0.38
Vitamin Premix ^a^	0.10	0.10
Mineral Premix ^b^	0.10	0.10
Choline CL 60%	0.003	0.00
Total	100	100
Calculated nutrient, %		
Metabolizable energy, kcal/kg	3000	3100
Dry matter	88.91	88.98
Crude protein	22.58	21.48
Crude fat	6.18	7.41
Crude fiber	2.55	2.49
Calcium	0.96	0.87
Non-phytate P	0.48	0.43
Digestible lysine	1.28	1.15
Digestible methionine	0.67	0.60
Digestible methionine and cysteine	0.95	0.87
Digestible threonine	0.86	0.77
Digestible arginine	1.32	1.26

^a^ Contains (in kg of vitamin premix): Vit. A = 2,400,000 IU; Vit. D = 1,000,000 IU; Vit. E = 16,000 IU; Vit. K = 800 mg; Vit. B1 = 600 mg; Vit. B2 = 1600 mg; Vit. B3 = 8000 mg; Vit. B5 = 3000 mg; Vit. B6 = 1000 mg; Vit. B7 = 40 mg; Vit. B9 = 400 mg; Vit. B12 = 6 mg. ^b^ Contains in kg of mineral premix: Cu = 2000 mg; Fe = 1200 mg; Mn = 18,000 mg; Se = 60 mg; Zn = 14,000 mg; I = 400 mg; Co = 80 mg.

**Table 3 animals-13-03024-t003:** Effect of gum arabic supplementation and stocking density on growth parameters of broiler chickens.

Item		Live Body Weight			Weight Gain			Feed Intake		
		1 d	14 d	36 d	1–14 d	15–36 d	1–36 d	1–14 d	15–36 d	1–36 d
TRT ^1^	SD ^2^									
CONT	Normal	39.17	397 ^b^	2269	358 ^b^	1871	2229	457 ^b^	2934	3392
T1		39.19	413 ^a^	2477	374 ^a^	2064	2438	471 ^a^	3017	3489
T2		39.17	391 ^b^	2413	352 ^b^	2022	2374	445 ^b^	2928	3373
T3		39.21	400 ^b^	2460	361 ^b^	2061	2421	454 ^b^	2976	3429
CONT	High	39.19	415 ^a^	2150	376 ^a^	1735	2111	446 ^b^	2822	3268
T1		39.20	399 ^b^	2263	360 ^b^	1864	2224	425 ^c^	2847	3272
T2		39.19	425 ^a^	2318	386 ^a^	1893	2279	464 ^ab^	2845	3308
T3		39.20	416 ^a^	2317	377 ^a^	1900	2277	445 ^bc^	2837	3282
SEM ^3^		0.02	7.08	28.72	7.12	26.60	28.70	8.90	27.64	31.74
SD										
Normal		39.19	400	2405 ^a^	361	2005 ^a^	2366 ^a^	456	2964 ^a^	3421 ^a^
High		39.20	414	2262 ^b^	374	1848 ^b^	2223 ^b^	445	2838 ^b^	3283 ^b^
SEM ^3^		0.01	3.54	14.36	3.56	13.30	14.35	4.45	13.82	15.87
TRT										
CONT		39.18	406	2209 ^b^	366	1803 ^b^	2170 ^b^	452	2878	3330
T1		39.20	406	2370 ^a^	367	1964 ^a^	2331 ^a^	448	2932	3381
T2		39.18	408	2365 ^a^	369	1958 ^a^	2326 ^a^	454	2887	3341
T3		39.20	408	2388 ^a^	369	1981 ^a^	2349 ^a^	449	2906	3356
SEM ^3^		0.01	5.01	20.31	5.03	18.81	20.29	6.29	19.54	22.44
Source of variance (*p*-Value)										
SD × TRT		0.864	0.014	0.2061	0.014	0.552	0.207	0.008	0.448	0.137
SD		0.350	0.012	<0.001	0.012	<0.001	<0.001	0.071	<0.001	<0.001
TRT		0.604	0.980	<0.001	0.977	<0.001	<0.001	0.917	0.232	0.428

^a–c^ Superscripts above the means for each parameter within column express the significant difference (*p* < 0.05). ^1^ Dietary treatments (TRT): CONT = the basal diet without supplement, T1 = the basal diets supplemented with 0.12% gum arabic, T2 = the basal diets supplemented with 0.25% gum arabic, and T3 = the basal diets supplemented with 0.10% inactivated stabilized *Saccharomyces cerevisiae* as commercial prebiotic. ^2^ Stocking density (SD): normal stocking density = 28 kg/m^2^ (12 chickens/pen), and high stocking density = 50 kg/m^2^ (22 chickens/pens). ^3^ SEM = Standard error of mean.

**Table 4 animals-13-03024-t004:** Effect of gum arabic supplementation and stocking density on feed efficiency and production of broiler chickens.

Item		Feed Conversion Ratio			Production Efficiency Index		
		1–14 d	15–36 d	1–36 d	1–14 d	15–36 d	1–36 d
TRT ^1^	SD ^2^						
CONT	Normal	1.28	1.56	1.52	221.7	402.2	414.5
T1		1.26	1.46	1.43	234.8	470.7	481.0
T2		1.26	1.45	1.43	221.5	462.8	471.8
T3		1.26	1.44	1.42	227.3	473.5	483.0
CONT	High	1.19	1.63	1.55	249.5	367.5	386.0
T1		1.18	1.53	1.47	241.7	411.7	427.2
T2		1.20	1.50	1.45	252.3	428.3	443.5
T3		1.18	1.49	1.44	252.2	431.0	446.5
SEM ^3^		0.016	0.014	0.012	6.01	8.27	8.22
SD							
Normal		1.26 ^a^	1.48 ^b^	1.45 ^b^	226.3 ^b^	452.2 ^a^	462.5 ^a^
High		1.19 ^b^	1.54 ^a^	1.48 ^a^	248.9 ^a^	409.6 ^b^	425.7 ^b^
SEM ^3^		0.008	0.007	0.006	3.00	4.13	4.11
TRT							
CONT		1.23	1.60 ^a^	1.53 ^a^	235.6	384.8 ^b^	400.2 ^b^
T1		1.22	1.50 ^b^	1.45 ^b^	238.2	441.2 ^a^	454.1 ^a^
T2		1.23	1.48 ^b^	1.44 ^b^	236.9	445.6 ^a^	457.7 ^a^
T3		1.22	1.47 ^b^	1.43 ^b^	239.7	452.2 ^a^	464.7 ^a^
SEM ^3^		0.011	0.009	0.008	4.25	5.84	5.81
Source of variance (*p*-Value)							
SD × TRT		0.799	0.967	0.965	0.205	0.418	0.377
SD		<0.001	<0.001	0.001	<0.001	<0.001	<0.001
TRT		0.804	<0.001	<0.001	0.911	<0.001	<0.001

^a,b^ Superscripts above the means for each parameter within column express the significant difference (*p* < 0.05). ^1^ Dietary treatments (TRT): CONT = the basal diet without supplement, T1 = the basal diets supplemented with 0.12% gum arabic, T2 = the basal diets supplemented with 0.25% gum arabic, and T3 = the basal diets supplemented with 0.10% inactivated stabilized *Saccharomyces cerevisiae* as commercial prebiotic. ^2^ Stocking density (SD): normal stocking density = 28 kg/m^2^ (12 chickens/pen), and high stocking density = 50 kg/m^2^ (22 chickens/pens). ^3^ SEM = Standard error of mean.

**Table 5 animals-13-03024-t005:** Effect of gum arabic supplementation and stocking density on relative small intestine morphology of broiler chickens.

Item		Weight ^4^			Length ^4^				
		Doud.	Jej.	Ile.	Total SI	Doud.	Jej.	Ile.	W:L
TRT ^1^	SD ^2^								
CONT	Normal	21.70	42.31	35.98	2.96 ^b^	16.48	42.91	40.59	0.37 ^b^
T1		22.24	40.37	37.38	4.06 ^a^	16.08	41.58	42.33	0.50 ^a^
T2		21.29	40.17	38.53	3.81 ^a^	16.80	41.91	41.28	0.48 ^a^
T3		20.52	37.88	41.58	3.59 ^a^	16.78	40.16	43.05	0.46 ^a^
CONT	High	20.80	42.44	36.75	2.93 ^b^	17.10	41.09	41.80	0.37 ^b^
T1		22.87	38.88	38.24	2.99 ^b^	17.18	41.07	41.73	0.35 ^b^
T2		21.61	38.94	39.44	3.07 ^b^	16.00	40.77	43.22	0.36 ^b^
T3		21.88	40.14	37.96	2.83 ^b^	16.29	42.08	41.62	0.34 ^b^
SEM ^3^		1.31	1.74	2.09	0.16	0.73	0.70	0.75	0.02
SD									
Normal		21.43	40.18	38.37	3.60	16.53	41.64	41.81	0.46
High		21.79	40.10	38.09	2.95	16.64	41.25	42.09	0.36
SEM ^3^		0.65	0.87	1.04	0.08	0.37	0.35	0.37	0.01
TRT									
CONT		21.25	42.37	36.36	2.95	16.79	42.00	41.20	0.37
T1		22.55	39.63	37.81	3.53	16.63	41.32	42.03	0.43
T2		21.45	39.55	38.98	3.44	16.40	41.34	42.25	0.42
T3		21.20	39.01	39.77	3.21	16.53	41.12	42.33	0.40
SEM ^3^		0.92	1.23	1.48	0.11	0.52	0.49	0.53	0.02
Source of variance (*p*-Value)									
SD × TRT		0.855	0.698	0.638	0.025	0.527	0.060	0.110	0.007
SD		0.703	0.948	0.854	<0.001	0.839	0.440	0.600	<0.001
TRT		0.705	0.233	0.402	0.006	0.959	0.624	0.426	0.041

^a,b^ Superscripts above the means for each parameter within column express the significant difference (*p* < 0.05). ^1^ Dietary treatments (TRT): CONT = the basal diet without supplement, T1 = the basal diets supplemented with 0.12% gum arabic, T2 = the basal diets supplemented with 0.25% gum arabic, and T3 = the basal diets supplemented with 0.10% inactivated stabilized *Saccharomyces cerevisiae* as commercial prebiotic. ^2^ Stocking density (SD): normal stocking density = 28 kg/m^2^ (12 chickens/pen), and high stocking density = 50 kg/m^2^ (22 chickens/pens). ^3^ SEM = Standard error of mean. ^4^ Doud = duodenum; Jej = jejunum; Ile = ileum; SI = small intestine; W:L = weight-to-length ratio.

**Table 6 animals-13-03024-t006:** Effect of gum arabic supplementation and stocking density on carcass characteristics and lymphoid organs of broiler chickens.

Item		Parameters ^4^								
		PSW (g)	CW (g)	DY (%)	Breast (%)	Legs (%)	Back (%)	Thymus (%)	Bursa (%)	Spleen (%)
TRT ^1^	SD ^2^									
CONT	Normal	2597	1968	75.85	28.70	28.09	8.19	0.33	0.15	0.10
T1		2639	1992	75.47	28.99	27.74	8.73	0.41	0.19	0.09
T2		2631	1983	75.39	29.17	26.50	7.49	0.40	0.19	0.09
T3		2652	1974	74.42	29.39	27.76	8.03	0.38	0.19	0.10
CONT	High	2283	1667	73.02	30.46	28.67	7.88	0.45	0.21	0.10
T1		2371	1766	74.49	29.90	29.10	7.94	0.57	0.20	0.10
T2		2446	1811	74.01	29.95	28.22	8.17	0.46	0.18	0.08
T3		2436	1809	74.25	29.44	28.52	8.26	0.46	0.17	0.09
SEM ^3^		44.60	33.69	0.47	0.66	0.65	0.25	0.04	0.02	0.01
SD										
Normal		2630 ^a^	1979 ^a^	75.28 ^a^	29.06	27.52 ^b^	8.11 ^a^	0.38 ^b^	0.18	0.10
High		2384 ^b^	1763 ^b^	73.94 ^b^	29.94	28.63 ^a^	8.06 ^b^	0.48 ^a^	0.19	0.09
SEM ^3^		22.30	16.84	0.23	0.33	0.32	0.12	0.02	0.01	0.01
TRT										
CONT		2440	1817	74.43	29.58	28.38	8.04	0.39	0.18	0.10
T1		2505	1879	74.98	29.45	28.42	8.33	0.49	0.20	0.09
T2		2538	1897	74.69	29.56	27.36	7.83	0.43	0.18	0.08
T3		2544	1892	74.33	29.42	28.14	8.15	0.42	0.18	0.09
SEM ^3^		31.54	23.82	0.33	0.47	0.46	0.18	0.03	0.01	0.01
Source of variance (*p*-Value)										
SD × TRT		0.494	0.177	0.092	0.660	0.790	0.054	0.592	0.105	0.555
SD		<0.001	<0.001	0.001	0.069	0.023	0.001	0.001	0.367	0.426
TRT		0.093	0.088	0.454	0.976	0.426	0.135	0.135	0.682	0.428

^a,b^ Superscripts above the means for each parameter within column express the significant difference (*p* < 0.05). ^1^ Dietary treatments (TRT): CONT = the basal diet without supplement, T1 = the basal diets supplemented with 0.12% gum arabic, T2 = the basal diets supplemented with 0.25% gum arabic, and T3 = the basal diets supplemented with 0.10% inactivated stabilized *Saccharomyces cerevisiae* as commercial prebiotic. ^2^ Stocking density (SD): normal stocking density = 28 kg/m^2^ (12 chickens/pen), and high stocking density = 50 kg/m^2^ (22 chickens/pens). ^3^ SEM = Standard error of mean. ^4^ PSW = pre-slaughter weight; CW = carcass weight; DY = dressing yield. Body components and lymphoid organs were computed as a ratio to live weight.

**Table 7 animals-13-03024-t007:** Effect of gum arabic supplementation and stocking density on blood biochemistry indicators of broiler chickens.

Item		Parameters ^4^								
		TP (g/dL)	ALB (g/dL)	GLU (mg/dL)	TG (mg/dL)	CHO (mg/dL)	UA (mg/dL)	CREA (mg/dL)	ALT (U/L)	AST (U/L)
TRT ^1^	SD ^2^									
CONT	Normal	3.39	1.95 ^b^	169.6	93.4	103.85	4.53	1.09	12	40
T1		3.40	1.72 ^c^	152.3	99.0	97.10	4.57	0.93	16	39
T2		3.39	1.70 ^c^	158.3	100.6	87.03	4.55	1.05	13	41
T3		3.23	2.13 ^a^	152.8	92.3	70.81	4.26	0.75	14	42
CONT	High	3.31	1.87 ^b^	155.2	105.0	78.95	4.94	1.02	15	57
T1		3.51	2.02 ^a^	163.6	110.4	96.91	4.64	1.17	16	49
T2		3.63	2.06 ^a^	150.7	106.7	91.95	4.80	1.25	15	45
T3		3.51	1.62 ^c^	154.8	96.5	92.98	5.03	1.28	14	51
SEM ^3^		0.17	0.11	8.70	9.79	6.92	0.17	0.12	2.03	3.23
SD										
Normal		3.35	1.88	158.2	96.3	89.70	4.48 ^b^	0.95 ^b^	14	41 ^b^
High		3.49	1.89	156.1	104.6	90.20	4.85 ^a^	1.18 ^a^	15	51 ^a^
SEM ^3^		0.09	0.05	4.35	4.89	3.46	0.09	0.06	1.01	1.61
TRT										
CONT		3.35	1.91	162.4	99.2	91.40	4.73	1.05	14	49
T1		3.45	1.87	158.0	104.7	97.01	4.60	1.05	16	45
T2		3.51	1.89	154.5	103.7	89.49	4.67	1.153	14	44
T3		3.37	1.89	153.8	94.4	81.90	4.65	1.01	15	47
SEM ^3^		0.12	0.08	6.15	6.92	4.89	0.12	0.08	1.43	2.28
Source of variance (*p*-Value)										
SD × TRT		0.732	0.001	0.486	0.974	0.015	0.232	0.095	0.789	0.288
SD		0.270	0.847	0.730	0.238	0.919	0.003	0.008	0.401	0.001
TRT		0.786	0.983	0.748	0.709	0.200	0.894	0.668	0.702	0.398

^a–c^ Superscripts above the means for each parameter within column express the significant difference (*p* < 0.05). ^1^ Dietary treatments (TRT): CONT = the basal diet without supplement, T1 = the basal diets supplemented with 0.12% gum arabic, T2 = the basal diets supplemented with 0.25% gum arabic, and T3 = the basal diets supplemented with 0.10% inactivated stabilized *Saccharomyces cerevisiae* as commercial prebiotic. ^2^ Stocking density (SD): normal stocking density = 28 kg/m^2^ (12 chickens/pen), and high stocking density = 50 kg/m^2^ (22 chickens/pens). ^3^ SEM = Standard error of mean. ^4^ TP = total protein, ALB = albumin, GLU = glucose, TG = triglycerides, CHO = cholesterol, UA = uric acid, CREA = creatinine, ALT = alanine aminotransferase, and AST = aspartate aminotransferase.

## Data Availability

All data presented in this study are available by all authors.

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
