# Peer review of "Evaluation of the Interaction between Gum Arabic Addition and Stocking Density on Growth Performance, Carcass Characteristics, and General Health Parameters of Broiler Chickens"

_animals, 2023, doi:10.3390/ani13193024_

Round 1

Reviewer 1 Report

The paper presented to me for evaluation addresses issues related to the use of gum arabic in reducing the stress of high stocking density. The work is interesting, but needs to be revised.
Material and Methods:
Please provide the species name of the yeast in italics (Line 86)
There is a formatting error in line 97
Why was 24 light program used ? This is not standard procedure.
In this section, please also indicate why a stocking density of 28 kg/m2 was used as the standard. Are there relevant regulations? If so please provide them.
Classically, feed and birds are weighed at 7-day periods or at feed changes. Why were other periods chosen? This is illogical to me.

Author Response

Date: 9 September, 2023‎

Dear Reviewer 1, ‎

On behalf of my colleagues, we would like to thank you for considering our ‎‎manuscript entitled "Evaluating the‎ Interaction Effect of Gum Arabic Supplementation and Stocking Density ‎on Growth Performance, Blood In-‎dices, Carcass Characteristics and Small Intestine ‎Morphology ‎of Broiler Chickens". ‎

We have revised the manuscript based on the reviewers’ comments point-by point ‎and ‎our responses to the reviewer are marked up using the “Track Changes” each ‎specific ‎comment. ‎

We hope the changes made are satisfactory to Your Excellency and to the ‎respected ‎reviewers. ‎

Best regards! ‎

Dr. Hani H. Al-Baadani

Dr. Rashed A. Alhotan

Q1: The work is interesting, but needs to be revised.

Authors’ Response: Thanks so much, I appreciate your efforts in your ‎valuable comments and question, which gave me the opportunity to improve manuscript.

Please if there are any opinions, guide us to correct it.

Q2: Please provide the species name of the yeast in italics (Line 86).‎

Authors’ Response: Done as requested in all manuscript.

Q3: There is a formatting error in line 97.

Authors’ Response: Thanks for your feedback. Done as requested in line 123-124.

Q4: Why was 24 light program used ? This is not standard procedure.

Authors’ Response: Thanks so much for your notification on this point. ‎It has been modified correctly in lines 130-132.

Q5: In this section, please also indicate why a stocking density of 28 kg/m2 was used as the standard. Are there relevant regulations? If so please provide them.

Authors’ Response: Thanks so much for your notification. We have revised in lines 119-120.

Q6: Classically, feed and birds are weighed at 7-day periods or at feed changes. Why were other periods chosen? This is illogical to me. ‎

Authors’ Response: Performance Evaluation including birds weight and feed weight were performed after feed change at the end of the starter (14 days) and grower stage (35 days)‎. Therefore, these periods chosen according to each of the feeding stages, which are mostly used commercially in the Kingdom of Saudi Arabia.

The manuscript has been completely revised, with some language changes made and ‎improved from our point of view, on the other hand, I appreciate your efforts in your ‎valuable comments and question, which gave me the opportunity to improve the ‎throughout manuscript.

Please if there are any opinions, guide us to correct it. ‎

satisfactory to you ‎

Thanks so much for your efforts. Your feedbacks are very valuable and will improve ‎my research skills and biological insight on my future studies.

--

Dr. Hani H. Al-Baadani

Dr. Rashed A. Alhotan

Reviewer 2 Report

The manuscript "Evaluating the Interaction Effect of Gum Arabic Supplementation and Stocking Density on Growth Performance, Blood Indices, Carcass Characteristics and Small Intestine Morphology of Broiler Chickens, was reviewed

The study was carried out properly, however, the main flaw is that the microbiota was not evaluated since gum Arabic and yest cell wall-derived products have a lot of impact on intestinal ecology. 

The criteria for blocking was not given. The use of blocks was not justified inside a very controlled environment. The block should be removed from analysis and the data should be re-analyzed. 

Another flaw is that the effect of increasing levels of gum Arabic was not evaluated, through orthogonal contrasts or analysis of variance,

Another mistake found is in the presentation of results, since when the interaction is statistically significant, the main effects lose validity, instead, in this study they were taken into account.

The Discussion is quite poor. At the beginning of the discussion of a group of response variables, it is not required to justify the evaluated response variables; this has no meaning.

In the discussion it is said that from 1-14 days, the higher stocking density had a positive effect on performance because it caused better thermoneutral conditions, which indicates that the chickens with low stocking density did not have adequate thermoneutral conditions. If the temperature of the building was tightly regulated, this suggests a fault in the actual temperature the chicks received. Which indicates another flaw in the management of the experiment.

For all these reasons, it is with great regret that I decide to reject the manuscript in its present form.

Extensive editing of English language required

Author Response

Date: 9 September, 2023‎

Dear Reviewer 2, ‎

On behalf of my colleagues, we would like to thank you for considering our ‎‎manuscript entitled "Evaluating the‎ Interaction Effect of Gum Arabic Supplementation and Stocking Density ‎on Growth Performance, Blood In-‎dices, Carcass Characteristics and Small Intestine ‎Morphology ‎of Broiler Chickens". ‎

We have revised the manuscript based on the reviewers’ comments point-by point ‎and ‎our responses to the reviewer are marked up using the “Track Changes” each ‎specific ‎comment. ‎

We hope the changes made are satisfactory to Your Excellency and to the ‎respected ‎reviewers. ‎

Best regards! ‎

Dr. Hani H. Al-Baadani

Dr. Rashed A. Alhotan

Q1: The study was carried out properly, however, the main flaw is that the microbiota was not evaluated since gum Arabic and yest cell wall-derived products have a lot of impact on intestinal ecology.‎

Authors’ Response: Thanks so much for your notification. This is part of a project where we evaluated growth performance, processing performance, intestinal morphology and blood biochemistry in the current paper and cecal microbiota + other related parameters in a future paper that will be submitted later. I agree with the reviewer it is informative to add such data to the current paper but sometimes we cannot answer all the research questions at once.

Q2: The criteria for blocking was not given. The use of blocks was not justified inside a very controlled environment. The block should be removed from analysis and the data should be re-analyzed.

Authors’ Response: Thanks so much for your notification. The block design was used because the enclosed housing system in which the study was conducted relied on negative pressure for the cooling and ventilation process while it was cold outside due to the local climate. Heat sources were also used to keep the chicks warm.

It is worth noting that the experiment was conducted in the winter in one building, following the same temperature schedule with slight temperature differences between opposite sides of the room. However, under practical conditions, it is sometimes difficult to achieve the exact temperature around the birds, under practical conditions unlike the room temperature. However, the use of blocks was justified inside study environment was justified to exclude any effect on the parameters studied.‎

Q3: Another flaw is that the effect of increasing levels of gum Arabic was not evaluated, through orthogonal contrasts or analysis of variance.

Authors’ Response: Thanks for your guidance. Tukey's test (p < ‎‎0.05) was used to detect significant differences between means of treatments. Therefore, the aim of this study was to evaluating the‎ interaction effect of gum Arabic and prebiotic ‎with various different stocking density conditions. Two way analysis of variance

If no interaction effects were observed between all dietary treatments on all the studied parameters that were evaluated, then we can resort to displaying the treatment effect only with orthogonal contrasts here.‎

Q4: Another mistake found is in the presentation of results, since when the interaction is statistically significant, the main effects lose validity, instead, in this study they were taken into account.

Authors’ Response: Thank you for pointing out that. Done as requested throughout the text (all manuscript).

Q5: The Discussion is quite poor. At the beginning of the discussion of a group of response variables, it is not required to justify the evaluated response variables; this has no meaning.

Authors’ Response: Thanks for your feedback. Done as requested throughout remove justify the evaluated response variables.

Q6: In the discussion it is said that from 1-14 days, the higher stocking density had a positive effect on performance because it caused better thermoneutral conditions, which indicates that the chickens with low stocking density did not have adequate thermoneutral conditions. If the temperature of the building was tightly regulated, this suggests a fault in the actual temperature the chicks received. Which indicates another flaw in the management of the experiment? ‎

Authors’ Response: The high SD group consumed numerically less feed, grew significantly better and had less FCR than the normal SD group at 14 days of age. These observations may be explained by the fact that chicks raised in cool weather at high stocking density produce less heat to keep themselves warm due to the microclimate created by crowding, maximizing feed utilization into meat. At such younger ages, chicks are not fully feathered, and when the environmental temperature is below the bird’s critical temperature, crowding becomes beneficial in lowering heat production. It is worthwhile to note that the trial was conducted in winter in one building, following the same temperature schedule with slight temperature variations between the opposite sides of the room. However, sometimes, it is difficult to reach the set temperature under practical conditions.

The manuscript has been completely revised, with some language changes made and ‎improved from our point of view, on the other hand, I appreciate your efforts in your ‎valuable comments and question, which gave me the opportunity to improve the ‎throughout manuscript.

Please if there are any opinions, guide us to correct it. ‎

satisfactory to you ‎

Thanks so much for your efforts. Your feedbacks are very valuable and will improve ‎my research skills and biological insight on my future studies.

--

Dr. Hani H. Al-Baadani

Dr. Rashed A. Alhotan

Reviewer 3 Report

Congratulations to the authors for the research, however, effects of prebiotics are well known in poultry science. For me, in the discussion it was not clear what is the knowledge gap that exists that justifies this research; as well as what are the hypotheses and technical justification for this research. In addition, authors should pay attention to terminologies and the use of appropriate technical terms. I will make a list of points that authors should revise, adjust or add to the manuscript.

1) long title, and which misuses the word "supplementation". Do the authors know what a supplement is? it's providing something more than what they are already eating; which is not the case here. Fit in the title and rest of the text.

2) introduction - pay attention to the previous comments

3) the authors talk about the chemical composition of the test product in the text and present it in Table 1; but do not describe the analysis methodology.

4) Table 2 only has the calculated composition of the rations; it is recommended to analyze and present the real/analyzed chemical composition.

5) The authors did not make it clear why they did not carry out the 3rd phase of rearing the chickens, using a finishing diet. Why is the experiment finished after 37 days and the authors aim to evaluate carcass?

6) I understood, but initially I had difficulty understanding the results presented in tables 4-8. Cleaner tables always attract more readers and increase the chance of your article being read. Transferring the main results to figures can be a good alternative and a way to make tables easier.

7) I enjoyed your discussion, but felt that the conclusion really lacked a conclusion. Remember that the conclusion must answer the objectives, using the interpretation of the results and in a relative language.

Author Response

Date: 9 September, 2023‎

Dear Reviewer 3, ‎

On behalf of my colleagues, we would like to thank you for considering our ‎‎manuscript entitled "Evaluating the‎ Interaction Effect of Gum Arabic Supplementation and Stocking Density ‎on Growth Performance, Blood In-‎dices, Carcass Characteristics and Small Intestine ‎Morphology ‎of Broiler Chickens". ‎

We have revised the manuscript based on the reviewers’ comments point-by point ‎and ‎our responses to the reviewer are marked up using the “Track Changes” each ‎specific ‎comment. ‎

We hope the changes made are satisfactory to Your Excellency and to the ‎respected ‎reviewers. ‎

Best regards! ‎

Dr. Hani H. Al-Baadani

Dr. Rashed A. Alhotan

Q1: In the discussion it was not clear what is the knowledge gap that exists that justifies this research; as well as what are the hypotheses and technical justification for this research. ‎

Authors’ Response: Thanks for your comment. Done as requested in lines 344-350.

Q2: long title, and which misuses the word "supplementation". Do the authors know what a supplement is? it's providing something more than what they are already eating; which is not the case here. Fit in the title and rest of the text.‎

Authors’ Response: Thank you for pointing out that. Done as requested, yes, supplementation is providing something more than Basal diet components eating. In other words, gum Arabic and prebiotic were used as supplement (feed additive) in the basal diet.

Q3: introduction - pay attention to the previous comments (hypotheses for this research).

Authors’ Response: Thanks for your feedback. Done as requested in introduction.

Q4: the authors talk about the chemical composition of the test product in the text and present it in Table 1; but do not describe the analysis methodology.

Authors’ Response: Done as requested in lines 96-102.

Q5: Table 2 only has the calculated composition of the rations; it is recommended to analyze and present the real/analyzed chemical composition. ‎

Authors’ Response: As a result of updating feed formulation programs through our continuous nutrient analysis for each feed ingredient prior to formulating the basal diet. Therefore, we do not find much change between the nutrient analysis and the calculated nutrients as we have done many previous studies.

Thank you for your comment, but in the future we are ready to evaluate the nutrient analysis to make the results more clear.

Q6: The authors did not make it clear why they did not carry out the 3rd phase of rearing the chickens, using a finishing diet. Why is the experiment finished after 37 days and the authors aim to evaluate carcass? ‎

Authors’ Response: The broilers in this study are slaughtered according to the usual requirements in the local market, as the appropriate slaughter weight (carcass) that is marketable is 1200 to 1800 kilograms.

Q7: I understood, but initially I had difficulty understanding the results presented in tables 4-8. Cleaner tables always attract more readers and increase the chance of your article being read. Transferring the main results to figures can be a good alternative and a way to make tables easier.

Authors’ Response: Thank you for your opinion. As a result of many study parameters and the study of interactions and main influences, we as authors believe that the presentation of results should be clearer and more accurate.

Q8: I enjoyed your discussion, but felt that the conclusion really lacked a conclusion. Remember that the conclusion must answer the objectives, using the interpretation of the results and in a relative language.

Authors’ Response: Done as requested as you indicated.

The manuscript has been completely revised, with some language changes made and ‎improved from our point of view, on the other hand, I appreciate your efforts in your ‎valuable comments and question, which gave me the opportunity to improve the ‎throughout manuscript.

Please if there are any opinions, guide us to correct it. ‎

satisfactory to you ‎

Thanks so much for your efforts. Your feedbacks are very valuable and will improve ‎my research skills and biological insight on my future studies.

--

Dr. Hani H. Al-Baadani

Dr. Rashed A. Alhotan

Reviewer 4 Report

Dear Corresponding Author,

Please find the specific comments and reviewer recommendations listed line-by-line below.

Title - selected blood indices instead of blood indices; what does the small intestine mean? is it duodenum, jejunum or ileum?

Simple summary 
L14-15 - Please correct according to the previous comments.
L15 - various instead of different
Comment 1 - It is inconsistent, i.e., once the Authors mentioned that the gum Arabic as a prebiotic was used (L13-14), another time the gum Arabic and prebiotics improved sth (L16). The Authors should be more precise.

Abstract
Comment 2 - The aim of the study should mirror the Title.
L22 - Please add the hybrid name of the broilers.
Comment 5 - "All chicks were weighed and randomly assigned to six blocks as replicates with eight treatments per block (forty-eight pens)" this information should be included.
Comment 4 - The p-value should also appear in the abstract section, particularly when significant differences between groups are observed.

Introduction
L53 microbiota instead of microflora
L63-65 - The authors should be focused on the gum Arabic, not other prebiotics. Thus, this sentence should be removed.
Comment 5- In the introduction section, there is a lack of information from the available literature about the effect of Arabic gum added to poultry diets on the selected blood biochemical parameters, carcass traits, and gut morphometrical measurements.

Material and methods
L86 Latin names must be written using italic font. Please double-check the whole manuscript in this case.
L86 The reviewer suggests to assign two control groups, i.e., negative and positive. Negative control - without additives, positive control with S. cerevisiae.
Comment 6 - There is no room to use the commercial names of the prebiotic preparations, such as Thepax. Use only S. cerevisiae prebiotic preparation in the whole manuscript. It is not an advertisement.
L97 - Why do the authors use an older version of the nutrient requirements for Ross 308? The latest is published in 2022.
L103 - Table 3 is not necessary and should be removed.
Comment 7 - There is no information about the experimental diet production process. Additionally, there is no info about the addition of feed additives, such as coccidiostats, exogenous enzymes, etc.
Comment 8 - Equations for the commonly used growth performance parameters are unnecessary and can be removed. 
L131 serum biochemical indices instead of blood indices.
L153 - It is not necessary, please remove
L156 - as above
L161 - as above
L164 - as above
Statistical analysis - L167
Comment 9 - How did the Authors calculate the homogeneity of variance? 

Recommendation - Please, consider changing the treatment abbreviations to, e.g., PC, NC, AG012, AG025. It makes the manuscript more readable and transparent.

Results

Comment 10 - The exact p-values should appear in the text, not only p < or > 0.05.
Comment 11  - From the reviewer's point of view, the results should be presented as follows: first, the interaction (if significant) rather than differences between treatments; if there is no significant interaction, the main effects should be presented. Thus, it is not necessary that dietary treatment response was not affected when the interaction between experimental factors is significant. Thus, the whole Results section should be rewritten according to the reviewer's suggestion.
Comment 12 - The order of the presented results should be changed as follows: growth performance, morphometric measurements of the intestine segments, carcass and immune organ index, and eat the end serum parameters.
Comment 13 - The tendency should be defined in the statistical analyses section because of the length of the jejunum (p = 0.06) and back (p = 0.054).

Discussion
Comment 14 - The explanation of the growth performance results should be connected with the morphometrics of the small intestine.

Conclusions
Comment 16 - There is a lack of statement that the gum Arabic affects the live body weight, weight gain, feed conversion ratio, and production efficiency index to the same extent as S. cerevisiae based on the entire experimental period results (1-36 d).

Author Response

Date: 9 September, 2023‎

Dear Reviewer 4, ‎

On behalf of my colleagues, we would like to thank you for considering our ‎‎manuscript entitled "Evaluating the‎ Interaction Effect of Gum Arabic Supplementation and Stocking Density ‎on Growth Performance, Blood In-‎dices, Carcass Characteristics and Small Intestine ‎Morphology ‎of Broiler Chickens". ‎

We have revised the manuscript based on the reviewers’ comments point-by point ‎and ‎our responses to the reviewer are marked up using the “Track Changes” each ‎specific ‎comment. ‎

We hope the changes made are satisfactory to Your Excellency and to the ‎respected ‎reviewers. ‎

Best regards! ‎

Dr. Hani H. Al-Baadani

Dr. Rashed A. Alhotan

Q1: Title - selected blood indices instead of blood indices. ‎

Authors’ Response: Thank you for pointing out that. Done as requested in line 4 and throughout the text (all manuscript).

Q2: Title - what does the small intestine mean? is it duodenum, jejunum or ileum? ‎

Authors’ Response: The small intestine includes the duodenum, jejunum, and ileum. It is indicated in line 160-161.

Q3: Simple summary: L14-15 - Please correct according to the previous comments.

Authors’ Response: Thanks for your feedback. Done as requested.

Q4: Simple summary: L15 - various instead of different.

Authors’ Response: Thank you for pointing. Done as requested in line 15 and all manuscript.

Q5: Simple summary: It is inconsistent, i.e., once the Authors mentioned that the gum Arabic as a prebiotic was used (L13-14), another time the gum Arabic and prebiotics improved sth (L16). The Authors should be more precise. ‎

Authors’ Response: Thanks for your feedback. Done as requested.

Q6: Abstract: The aim of the study should mirror the Title.‎

Authors’ Response: Done as requested.

Q7: Abstract: L22 - Please add the hybrid name of the broilers.

Authors’ Response: Done as requested in line 26.

Q8: Abstract: "All chicks were weighed and randomly assigned to six blocks as replicates with eight treatments per block (forty-eight pens)" this information should be included.

Authors’ Response: Done as requested in lines 26-27.

Q9: Abstract: The p-value should also appear in the abstract section, particularly when significant differences between groups are observed.

Authors’ Response: Done as requested in all Abstract.

Q10: Introduction: L53 microbiota instead of microflora. ‎

Authors’ Response: Done as requested in line 63.

Q11: Introduction: L63-65 - The authors should be focused on the gum Arabic, not other prebiotics. Thus, this sentence should be removed. ‎

Authors’ Response:  Done as requested as you indicated.

Q12: In the introduction section, there is a lack of information from the available literature about the effect of Arabic gum added to poultry diets on the selected blood biochemical parameters, carcass traits, and gut morphometrical measurements.

Authors’ Response: Done as requested throughout the text (all manuscript).

Q13: Material and methods: L86 Latin names must be written using italic font. Please double-check the whole manuscript in this case.

Authors’ Response: Done as requested throughout the text (all manuscript).

Q14: Material and methods: L86 The reviewer suggests to assign two control groups, i.e., negative and positive. Negative control - without additives, positive control with S. cerevisiae. ‎

Authors’ Response: Thank you very much for your efforts. The result of control groups (negative and positive) in this study is stocking density (normal and high) and therefore a new design is needed to confirm this good suggestion, so we will work hard by continuing future studies to conduct this comparison to evaluate the mechanism of action of gum Arabic in broiler chickens.

Q15: Material and methods: There is no room to use the commercial names of the prebiotic preparations, such as Thepax. Use only S. cerevisiae prebiotic preparation in the whole manuscript. It is not an advertisement. ‎

Authors’ Response: Thank you for pointing out that. Done as requested throughout the text (all manuscript).

Q16: Material and methods: L97 - Why do the authors use an older version of the nutrient requirements for Ross 308? The latest is published in 2022.

Authors’ Response: Aviagen has released new broiler performance objectives and nutrition specifications for Ross brands. The 2022 Broiler Nutrition Specifications are the result of internal and external research, literature reviews, and consideration of commercial nutrition practices but it does not differ significantly with the 2019 version. The big difference is when the requirements for broiler chickens are used by NRC.

In comparison to the previous Broiler Nutrition Specifications released in 2019, nutrints recommendation takes aligned with optimal genetic potential. 

Q17: Material and methods: L103 - Table 3 is not necessary and should be removed.

Authors’ Response:  Table 3 has been removed.

Q18: Material and methods: There is no information about the experimental diet production process. Additionally, there is no info about the addition of feed additives, such as coccidiostats, exogenous enzymes, etc.

Authors’ Response: The experimental diet is the basal diet used in all dietary treatments, where gum Arabic or prebiotic were used as additive to the basal diet in a more precise way in terms of gradual mixing, and in the end an automatic vertical mixer was used for the homogenization process.

We have indicated, as usual in studies, and according to the nutritional needs of the Ross strain used in this study in line 122 to 124, as well as the basic ration components used in Table 2 without addition of feed additives, such as coccidiostats, exogenous enzymes, etc.

Q19: Material and methods: Equations for the commonly used growth performance parameters are unnecessary and can be removed. ‎

Authors’ Response: Thank you very much for this suggestion. It is correct. All equations have been removed as they are also described in the references referred to.

Q20: Material and methods: L131 serum biochemical indices instead of blood indices.

Authors’ Response: Done as requested in line 181.

Q21: Material and methods: L153, L156, L161, L164- It is not necessary, please remove.

Authors’ Response: All equations have been removed.

Q22: Statistical analysis: L167- How did the Authors calculate the homogeneity of variance?.

Authors’ Response: Done as requested in line 200.

Q23: Recommendation - Please, consider changing the treatment abbreviations to, e.g., PC, NC, AG012, AG025. It makes the manuscript more readable and transparent.

Authors’ Response: Thank you very much for your notification. But we use it when we do a negative and positive control group with other treatments without looking at the interference. So it is a bit more complicated from our point of view. I hope if you have another opinion, guide us to that.

Q24: Results: The exact p-values should appear in the text, not only p < or > 0.05. ‎

Authors’ Response: Done as requested in all Results.

Q25: Results: From the reviewer's point of view, the results should be presented as follows: first, the interaction (if significant) rather than differences between treatments; if there is no significant interaction, the main effects should be presented. Thus, it is not necessary that dietary treatment response was not affected when the interaction between experimental factors is significant. Thus, the whole Results section should be rewritten according to the reviewer's suggestion.

Authors’ Response: Done as requested in Results.

Q26: Results: The order of the presented results should be changed as follows: growth performance, morphometric measurements of the intestine segments, carcass and immune organ index, and eat the end serum parameters.

Authors’ Response: Done as requested in Results.

Q27: Results: The tendency should be defined in the statistical analyses section because of the length of the jejunum (p = 0.06) and back (p = 0.054).

Authors’ Response: detect significant differences between means at  p < ‎‎0.05 as description in the statistical analyses section.

Q28: Discussion: The explanation of the growth performance results should be connected with the morphometrics of the small intestine. ‎

Authors’ Response: Done as requested in lines 384-395.

Q29: Conclusions: There is a lack of statement that the gum Arabic affects the live body weight, weight gain, feed conversion ratio, and production efficiency index to the same extent as S. cerevisiae based on the entire experimental period results (1-36 d).

Authors’ Response: Done as requested.

The manuscript has been completely revised, with some language changes made and ‎improved from our point of view, on the other hand, I appreciate your efforts in your ‎valuable comments and question, which gave me the opportunity to improve the ‎throughout manuscript.

Please if there are any opinions, guide us to correct it. ‎

satisfactory to you ‎

Thanks so much for your efforts. Your feedbacks are very valuable and will improve ‎my research skills and biological insight on my future studies.

--

Dr. Hani H. Al-Baadani

Dr. Rashed A. Alhotan

Round 2

Reviewer 1 Report

The authors made all the corrections I raised or addressed ambiguities in their response to the review.

Therefore, I believe that the work can be accepted for further processing

Reviewer 3 Report

The requested adjustments were made. The text has become clearer, so I am in favor of publishing it.

Reviewer 4 Report

Dear Editor,

The manuscript (ID animals-2587295) titled "Evaluating the‎ Interaction Effect of Gum Arabic Supplementation and Stocking Density ‎on Growth Performance, Blood Indices, Carcass Characteristics and Small Intestine ‎Morphology of Broiler Chickens" will be accepted after minor revision.

Kind regards,
Section Board Editor